**communications** sustainability

# Comparing corporate sustainability programmes, social entrepreneurship, and cooperatives in shaping farmers' well-being

Javier G. Montoya-Zumaeta [1] ✉, Christoph Oberlack [1,2], Ronja Barelli [3], Samuel Bruelisauer [1,4] & Diego P. Zavaleta [5]

The governance of sustainability in agri-food value chains has evolved beyond conventional certification strategies. Corporate sustainability programs, social enterprises, and cooperatives are key complementary or alternative strategies. However, their relative contributions to farmers' well-being remain unclear. Therefore, we conducted a household survey of 634 cocoa and coffee farmers distributed across three production hotspots in the Peruvian Amazon. We use the Personal Well-being Index to compare well-being among and between farmers involved in the various strategies. Results show that farmers engaged in any strategy of sustainability governance report higher well-being than independent farmers. However, only social enterprise strategies remained a significant contributor to overall farmers' well-being after controlling for certification, context, household, and demographic variables. Our findings challenge traditional productivity centered approaches and emphasize the importance of incorporating localized, socially tailored strategies to enhance the well-being of farmers within agri-food value chains.

Increasing awareness about the role of food production for rural well-being and nature conservation has boosted the rise and evolution of diverse sustainability strategies embedded in value chains[1–3]. In particular, sustainability certifications such as Fairtrade have become a key strategy intended to enhance human well-being in cocoa, coffee, and other global agri-food value chains[4,5]. However, accumulating evidence points to limited – and sometimes even adverse – contributions of certification schemes to well-being[6–9]. Typical barriers include high adoption costs[10–12], flawed auditing practices[13], imbalanced market structures[14,15], and weak property rights[16]. Indeed, the limitations of sustainability certification are well known, but rigorous understanding of the well-being contributions of other emerging sustainability strategies remains elusive.

Among these other strategies, corporate sustainability programmes, social entrepreneurship, and cooperative-based initiatives are particularly relevant[2,17,18]. Many large companies now operate corporate sustainability programmes[19]. These programmes are directly managed by companies – including traders, manufacturers, or retailers – for their own supply chains[1]. They typically involve company-defined sustainability goals, sourcing policies, impact-oriented projects, and systems for traceability and verification of sustainability claims[20]. By contrast, social enterprises and cooperatives promote social and solidarity economy strategies[21]. Social enterprises take a business approach to attain a social and environmental purpose[22]. They often aim to create change by prioritizing this corporate purpose over profit in decision-making, with the former enshrined in corporate mission and accountability structures. Fair prices, direct trade relations and risk sharing are key instruments of many social enterprises in the agri-food sector[23]. Meanwhile, strategies based on cooperatives claim democratic governance and co-ownership models as their distinctive features, enabling smallholders – independently or with the support of their partners – to seek greater access and control over their roles in agri-food value chains[19,21,24]. They offer producer-led financial profitability as well as social and cultural benefits to their members[25,26]. The common denominator of these governance strategies is that they attempt to influence the rules and norms that govern land use, investment, and trade in global agri-food value chains. They differ in terms of the main governing actors (corporations, social entrepreneurs, producers, standard setters) and the underlying

[1]Centre for Development and Environment (CDE), University of Bern, Bern, Switzerland. [2]Department of Social Sciences, University of Bern, Bern, Switzerland. [3]Department of Psychology, University of Zurich, Zurich, Switzerland. [4]Institute of Geography, University of Bern, Bern, Switzerland. [5]Alliance of Bioversity International and CIAT, Lima, Peru. ✉e-mail: javier.montoya@unibe.ch

mechanisms employed to promote well-being (e.g., standards; business models; projects; value capturing by different actors).

Existing evidence and theory on the pathways through which corporate programmes, social enterprises, and cooperative-based strategies influence well-being are inconsistent. Corporate programmes may contribute to farmers' well-being by offering them access to income opportunities, farm inputs, or professionalized services at scale[1,17]. However, some programmes undermine the livelihood diversification of smallholders, as seen in West Africa among farmers who were pressured to specialize in cocoa production[27]. Some political economists argue that corporate programmes are used to accumulate capital in the hands of multinational companies while squeezing the value of sustainable products from smallholders – even perpetuating neo-colonialist practices[28,29]. This can occur through coercive use of power, arising from oligopolistic value chains[30]. Meanwhile, many social enterprises and cooperative-based strategies present themselves as radical alternatives to agro-industrial food production, motivated by norms of solidarity, conservation, and agroecology[31]. They often seek to improve farmers' well-being through fair price mechanisms and risk sharing, among other ways[2]. Cooperatives, in particular, often surge from existing communal organizations and attempt to incorporate, disseminate, and strengthen collective values across their members as a pathway to enhance their well-being while maintaining roots with communities[25]. Even though these strategies have spread throughout agri-food value chains in recent years, they remain niche innovations in terms of overall production[32,33]. Barriers to scaling include limited access to finance, lack of public recognition, and inadequate legal and fiscal frameworks[22]. Further, social enterprises and cooperatives may also exhibit asymmetric wealth and power accumulation among elites within the organization, undermining the effectiveness and reach of their potential contributions to well-being[29].

Most research on corporate sustainability programmes, social enterprises, cooperatives, or certification tends to focus on one of these strategies to the exclusion of the others. There is a lack of comparative analyses that examine whether and how these strategies influence well-being relative to each other. This gap is highly problematic; the absence of comparative data and theory strongly limits our understanding of the relationships between these strategies and human well-being. It also precludes insights into the suitability of particular strategies to address specific social-ecological contexts and market conditions.

Against this background, the present analysis aims to compare the well-being of coffee and cocoa farmers engaged in corporate sustainability programmes, social enterprises, and cooperatives. To this end, we conducted a cross-sectional survey of 634 coffee and cocoa farmers in three sites across the Peruvian Amazon. We surveyed well-being, demographic features, land use practices, income sources, as well as respondents' assessment of their organizations' services and vulnerability to shocks. We used the Personal Well-being Index (PWI) to measure farmers' well-being[34]. The PWI measures participants' self-reported states in eight well-being domains, namely: living standard, health, life achievements, personal relationships, safety, community integration, future security, and spirituality. The domains are combined to construct a composite indicator (PWI-8). Use of the PWI enabled us to compare both overall well-being and the individual components of well-being among and between farmers involved in the different sustainability governance strategies. We also compared the PWI with a reference group of independent farmers who sold their production through local intermediaries without any sustainability governance strategy. Apart from the current state of PWI, the survey also asked farmers about their perception of possible changes in each PWI domain as a result of their participation in the sustainability strategies (corporate sustainability programme, social enterprise, or cooperative).

Our analysis focuses on three coffee and cocoa production hotspots in Peru (Fig. 1). Cocoa and coffee are among the most important agricultural commodities associated with persistent poverty as well as deforestation[35–40]. Despite recently soaring commodity prices, about 70–80% of cocoa and coffee farming families do not earn enough for a decent living[35,36]. Peru is an important producer of cocoa and coffee, placing eighth worldwide in terms

of production volumes of either commodity in 2023[41]. Peru is also a leader in organic production of these commodities[42]. Across its territory, around 313,000 farmers rely on at least one of these crops for their livelihoods, and most of them are settled in the Amazon[43] – a region of critical ecological relevance for the entire globe.

## Results

### Comparing overall well-being across sustainability governance strategies

First, our results indicated a significant difference of 6.9 ($p < 0.01$) points between the median score for the composite well-being index (PWI-8) of farmers participating in any of the assessed strategies (PWI-8$_{yes}$ = 85.6) versus those working independently (PWI-8$_{no}$ = 78.7), respectively (Fig. 2A). This result is supported by comparison of the PWI-8 median scores for each of the strategies, individually, with the median PWI-8 score of independent producers: independent farmers' PWI-8 scores were significantly lower than the those reported by farmers engaged in corporate sustainability programmes ($p < 0.01$), cooperatives ($p < 0.01$), and social enterprises ($p < 0.01$), respectively (Fig. 2B). No significant differences in PWI-8 median scores were detected between farmers producing for corporate programmes, social enterprises, or cooperatives (Fig. 2B).

### Disentangling well-being domains by sustainability strategy

Next, we examine how the individual strategies performed in each of the eight well-being domains of the PWI, as self-reported by participating farmers. As the results in Fig. 3 demonstrate, farmers engaged in corporate sustainability programmes reported significantly higher satisfaction in terms of their standard of living, personal relationships, and community integration, when compared to independent producers. However, they reported less satisfaction in terms of personal relationships, community integration, future security, and spirituality, when compared to farmers engaged in social enterprises or cooperative strategies.

Farmers engaged in cooperatives reported higher satisfaction than independent producers in most well-being domains, but especially in living standard, life achievements, personal relationships, safety, community integration, and spirituality. They are also more satisfied than participants in corporate programmes regarding their personal relationships, community integration, and spirituality. However, compared with participants in social enterprises, they report less satisfaction in terms of their future security.

Farmers engaged in social enterprises reported significantly higher satisfaction than independent producers in terms of living standards, personal relationships, community integration, future security, and spirituality. They also reported significantly higher satisfaction regarding future security compared to all other farmers. No significant differences between groups were detected in terms of self-reported satisfaction with individual health, nor between farmers engaged in any strategy in terms of self-reported standard of living.

The results indicate that, on average, self-reported farmer well-being improved for all domains across all groups, when compared to the period before engagement in one of the sustainability strategies (Figure S1 in the Supplementary Information). Most notably, in comparison with the other farmer groups, farmers engaged in social enterprises reported high levels of positive change and perceived contributions of the corresponding sustainability strategy for five well-being domains: standard of living, life achievements, safety, community integration, and future security. To a lesser extent, farmers engaged in corporate sustainability programmes and cooperative strategies, respectively, reported medium levels of positive change in living standards, life achievement, safety, and future security.

### The influence of demographic and social-ecological context factors

We performed multivariate regressions to test relationships between farmers' well-being and several variables capturing household, demographic, and context features. Further, we included a binary variable indicating whether farmers' main crop was certified under any sustainability

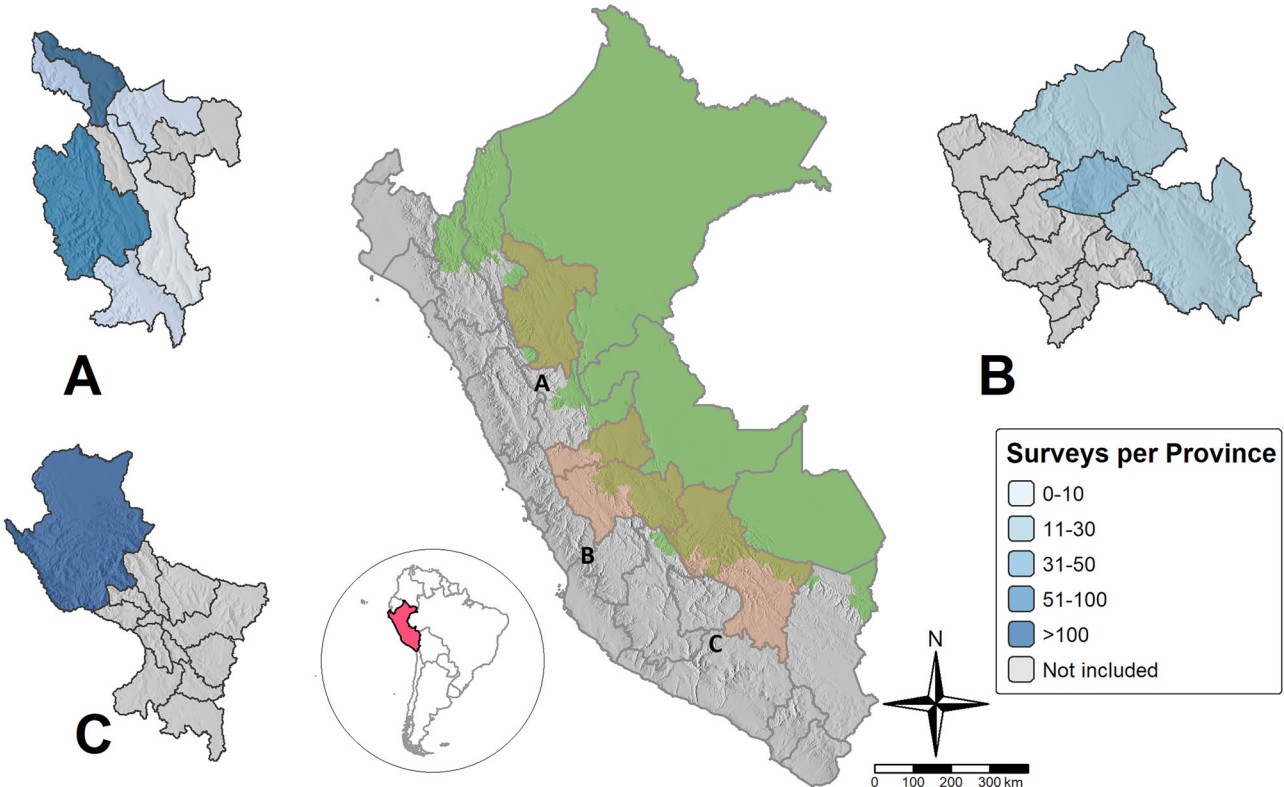

**Fig. 1 | Location of the three major coffee and cocoa production areas examined in our study. A** Area 1 - San Martin (SM), located within the San Martin department; (**B**) Area 2 - Selva Central (SC), covering the departments of Junín and Pasco; (**C**) Area 3 - Quillabamba (QU), located in the northern part of the Cusco department.

**Fig. 2 | Personal Well-being Index (PWI-8) comparisons across farmer groups.** Panel (**A**) shows the PWI-8 comparison between participants in any strategy (*n* = 537) versus farmers working independently (IND, *n* = 86). Panel (**B**) shows the pairwise PWI-8 comparisons of IND farmers with farmers engaged in corporate sustainability programmes (CSP, *n* = 254), cooperatives (CO, *n* = 214), and social enterprises (SE, *n* = 77), respectively. Statistical mean comparisons were performed using the Wilcoxon rank test with Holm correction for unpaired samples. ***, **, and * reflect statistical differences at 1%, 5%, and 10%, respectively.

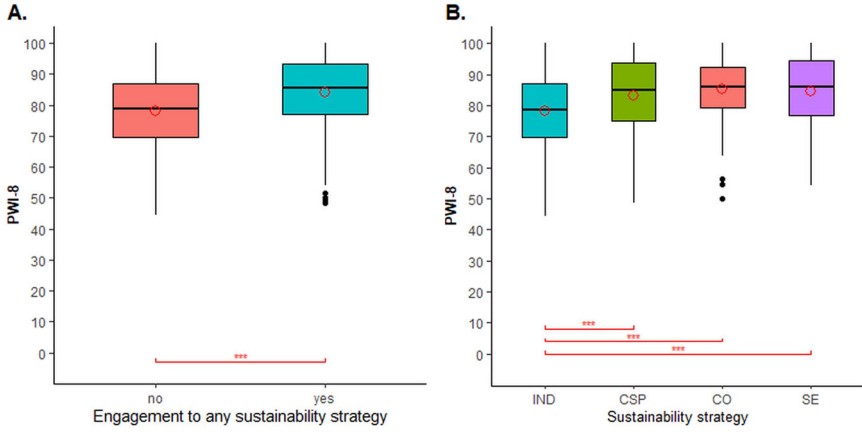

label in order to capture the possible contribution of certification to well-being. We opted for this approach given the presence of certified farmers in all farmer groups (Table S1–Supplementary Information). Our results indicate that social enterprises are the only strategy that maintains statistical significance for well-being status after controlling for certification as well as contextual, demographic, and household variables (Fig. 4). Farmers' engagement with social enterprises was associated with an average increase of 4.4 points in PWI-8 compared to independent farmers. Meanwhile, variables capturing farmer engagement in corporate sustainability programmes, cooperatives, or (main) crop certification showed positive but statistically non-significant relationships with well-being.

Having one additional hectare of land in the main crop (cocoa or coffee) was associated with an average increase of 0.6 points on the composite PWI-8. Conversely, variables significantly associated with reductions in PWI-8 score were of a contextual and demographic nature. Specifically,

farmers settled in the regions of Quillabamba and Selva Central displayed PWI-8 scores that were 5.8 and 9.5 points lower, respectively, when compared to farmers in the San Martin region. This reflects differences among these sites in terms of market accessibility, ecological and institutional factors, as we discuss further in the next Section. Female farmers displayed PWI-8 scores that were an average of 4.6 points lower than those of men. Further, a marginal increase (+10 years) in farmers' age was associated with a reduction of 1.6 PWI-8 points. Finally, PWI-8 scores were 3.9 points lower (*p* < 0.01) on average among farmers who sold more than 50% of their main crop to buyers other than the organization implementing the sustainability strategies they were engaged in.

**The role of farm size and main crop**

In order to test for heterogeneous within-group relationships between the assessed strategies and farmers' well-being, we performed additional

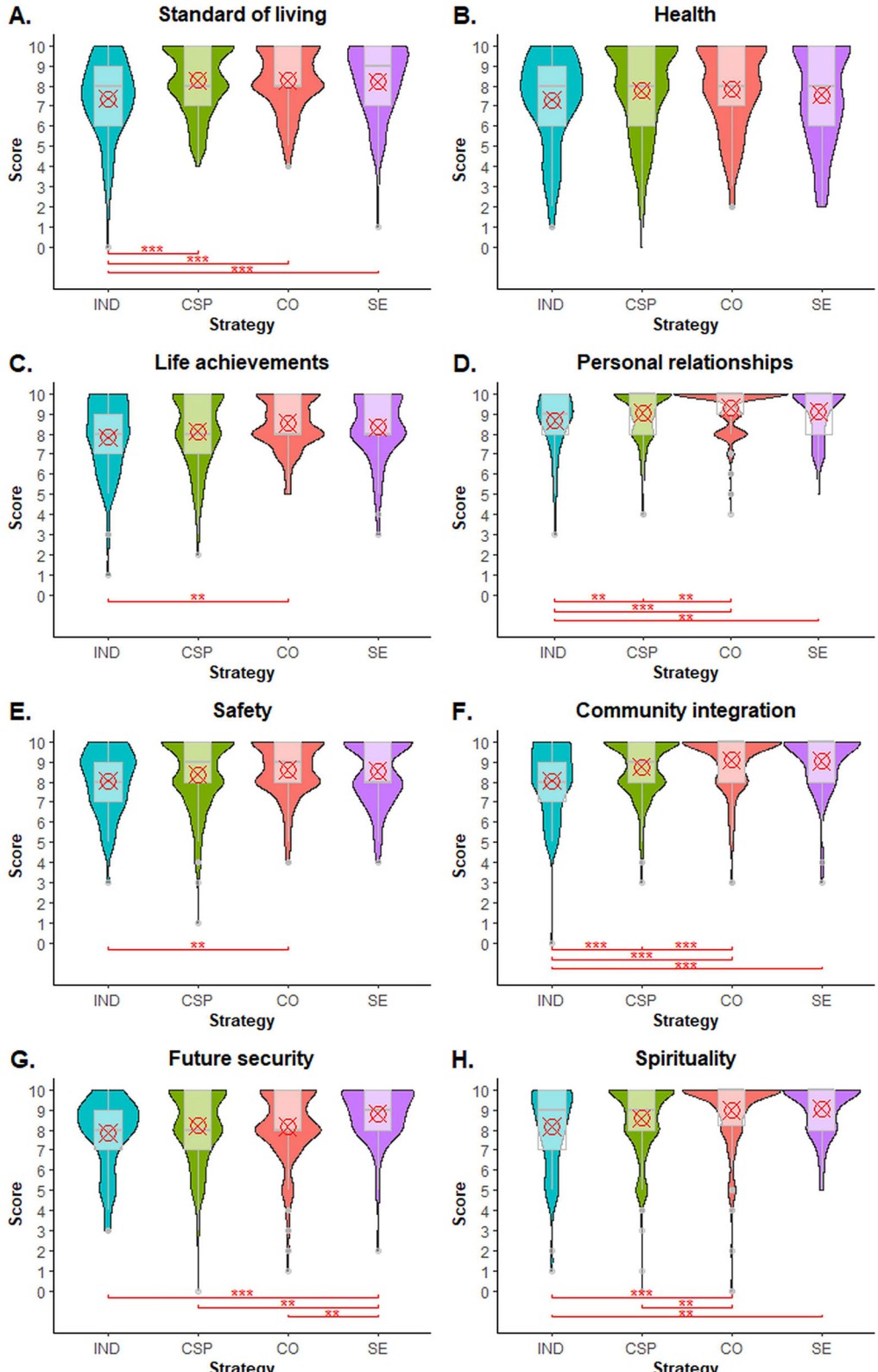

**Fig. 3 | Comparisons of individual domains of Personal Well-being Index (PWI) across sustainability strategies.** Comparison of farmers by sustainability strategy (IND independent, CSP corporate sustainability programme, CO cooperatives, SE social enterprise) for each self-reported PWI domain: standard of living (**A**), health (**B**), life achievements (**C**), personal relationships (**D**), safety (**E**), community integration (**F**), future security (**G**), and spirituality (**H**). Statistical median comparisons were performed using the Wilcoxon rank test for unpaired samples with Holm correction. ***, **, and * reflect statistical differences at 1%, 5%, and 10%, respectively.

**Fig. 4 | Factors influencing Personal Well-being Index (PWI) among coffee and cocoa farmers.** Regressions were performed on observations with complete data ($n = 599$). Marginal changes to well-being based on the overall sample are reported to facilitate interpretation. ***, **, and * reflect statistical significance at 1%, 5%, or 10%, respectively.

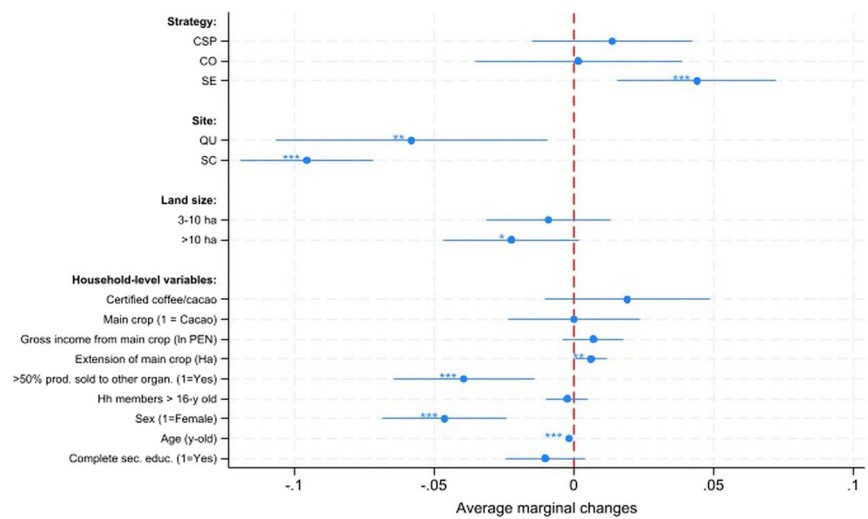

**Fig. 5 | Heterogeneity analysis.** Marginal changes in well-being among subsets of farmer groups are reported to facilitate interpretation. The results of regression analysis of farmer subgroups based on management of small ($n = 263$), medium ($n = 245$), or large ($n = 126$) farms are shown in the left panel; results of regression analysis of farmer subgroups based on cultivation of cocoa ($n = 382$) or coffee ($n = 252$) are shown in the right panel. ***, **, and * reflect statistical significance at 1%, 5%, or 10%, respectively.

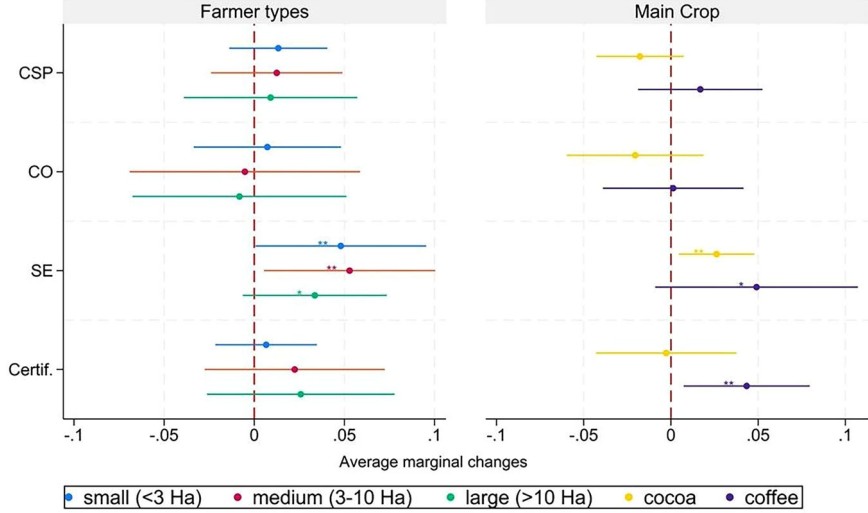

regression analyses of subsamples determined by farm size and main crop. Table S2 in the Supplementary Information displays the complete results of regressions on both the overall sample and specific farmer groups. Notably, engagement in social enterprises was associated with significantly higher well-being across all farmer groups, but at varied magnitudes (Fig. 5). In particular, the PWI-8 score of social-enterprise farmers with medium-sized farms (3–10 ha) was 5.3 points higher on average than that of the reference group comprised by independent farmers. Among social-enterprise farmers managing smaller (<3 ha) and larger (>10 ha) farms, average PWI-8 scores were 4.8 and 3.4 points higher than the comparison group, respectively. Similarly, social-enterprise farmers who mainly cultivated cocoa, on the one hand, or coffee, on the other, displayed average PWI-8 scores that were 2.6 or 4.9 points higher than the comparison group. Finally, we found that farmers cultivating certified coffee had PWI-8 scores that were 4.3 points higher on average than those whose coffee was not certified.

## Discussion
### Comparing the well-being contributions of corporate programmes, cooperatives, and social enterprises
Overall, we found that farmers engaged in corporate sustainability programmes, social enterprises, or cooperatives reported higher levels of well-being compared to those selling to conventional supply chains without sustainability governance. However, after controlling for the adoption of any sustainability certification (namely Organic, Fairtrade, or Rainforest Alliance) as well as for context, household, and demographic variables, only social enterprise strategies remained as a statistically significant contributor to overall well-being. There are several possible reasons for this, as it can be inferred from analysing the characteristics of cases included in each class of governance strategy (see Table S4 - Supplementary Information). The governance approach of many corporate sustainability programmes closely resembles that of certified supply chains, in that companies establish their own sustainability standards and provide compliant producers with price premiums, inputs, and support services[20]. Often, corporations utilize third-party certification in their own sustainability programmes as part of their offerings to their customers[20]. Similarly, many cooperatives were explicitly formed to implement certifications in the cocoa and coffee sectors[44]. Controlling for certification may, therefore, reduce the statistical significance attributed to corporate sustainability programmes or cooperatives.

On the other hand, the significant association found between social enterprises and farmers' well-being may stem from a combination of geographical and social targeting, purpose-driven organizational logics, business approach, and capacity for development, all of which are incorporated to some extent in their interventions (see Table S4-Supplementary Information). The social enterprises in our sample intervene in highly contextualized ways, consciously targeting specific geographical areas (e.g., around protected areas or communal forests) and specific farmer groups

(e.g., former coca – *Erythroxylum coca*–producers, indigenous communities, women). They often originate from, or maintain a long-term personal presence in, the cocoa- or coffee-origin area in which they operate. This deep local engagement enables them to implement tailored interventions. These interventions incorporate, for instance, emergency insurance, educational support for farmers' children, and strengthening of farmers capabilities in areas such as leadership, entrepreneurship, and gender equity. Interventions may be calibrated according to individual needs assessments and follow-ups by technical assistants, sometimes aided by tools such as soil analysis that make it possible to customize support to each farmer's context. While these types of services are not exclusive to social enterprises, the strong local embeddedness of these enterprises enables them to adapt their interventions to the specific challenges and needs of a given region[45]. This tailored, personalized approach appears to not only increase the efficacy of farmer support activities, but may also contribute to the higher sense of recognition farmers reported in association with social enterprises[46]. This recognition, in turn, appears to foster stronger social relationships among farmers, as demonstrated by survey responses in which farmers attributed higher contributions by social enterprises to community and personal relationships. Conversely, the corporate programmes in our sample are more likely to be designed in distant headquarters for large intervention areas spanning multiple countries. This more standardized approach may sacrifice the quality-of-service portfolios vis-à-vis the specific issues in a given region. Further, many corporate sustainability programmes emphasize interventions that prioritize yields and productivity, whereas social enterprises take more of a holistic business approach to sustainable development. Cooperatives are locally rooted organizations, too, but most of the cooperatives in our sample remain dependent on external project funding. This funding structure may constrain the economic viability of service delivery and customization[47]. It differs from social enterprises that adopt a clear business approach to sustainable development.

Another reason for the association between social enterprises and higher farmer well-being may relate to the value propositions pursued by different business models[33,48]. Most of the cooperatives and corporate programme participants in our sample act as midstream actors in value chains, supplying large volumes of coffee and cocoa to manufacturers and roasters in the upstream value chain[1]. By contrast, most of the social enterprises in our sample employ more sophisticated business models that enable farmers to capture a greater share of the value created along the value chain[49]. These innovations include, for example, diversification of outputs with higher quality standards and differentiated company values like inclusion and community empowerment. These social enterprises also incorporate more inclusive decision-making processes – often as a consequence of their targeted approach to farmer engagement.

In addition, our results show that satisfaction levels with living standards are comparable across all assessed strategies. In regard to future security, however, social-enterprise farmers again report higher satisfaction than other farmer groups. This may reflect a tendency among social enterprises to apply more holistic approaches to improve farmer well-being. This contrasts with the approaches of most of the corporate sustainability programmes and also a few of the large cooperatives selected for this study, which instead emphasize increasing farmer productivity as the pathway to improve well-being among their farmers[27,28]. Further, although farmers in cooperatives reported higher satisfaction with personal relationships and community integration than farmers in corporate programmes, the contribution of this strategy was, on average, lower than in social enterprises. This suggests a certain loss of emphasis on relationships among cooperatives when they strive to improve their financial profitability, frequently by enlarging their intervention areas to integrate new members [26].

### Sustainability governance in view of contextual and demographic differences

Well-being measures differed across study sites and demographic groups. Notably, farmers in our analyses reported, on average, a PWI-8 of 83.3. That average is higher than has been found in other studies of similar populations, with averages typically falling in the 70–80 range[34,50–52]. However, the relatively high score of our overall sample was mainly driven by the higher well-being reported by San Martin farmers (PWI-8$_{SM}$ = 86.9) when compared to responses from Selva Central (PWI-8$_{SC}$ = 75.8) and Quillabamba (PWI-8$_{QU}$ = 76.5) farmers. The high well-being levels reported in San Martin may stem from the superior access of its cocoa and coffee production areas to markets, its higher farm productivity, and the greater presence of intermediaries in the area who possess the capacity to exploit market conditions for the benefit of farmers. Further, our surveys were conducted in the second half of 2024 when international cocoa prices attained a historic peak, reaching almost five times the price of the previous year.

Differences in institutional conditions and market access may also explain why the social enterprises and cooperatives included in our analyses are largely clustered in San Martin as opposed to the other study sites[47]. Notably, this department currently ranks first nationwide in the production of both cocoa and coffee, thanks also to the efforts of multiple stakeholders working in the region, including NGOs and government programmes that promote initiatives for farmer empowerment as a pathway to sustainable development[53]. Notably, the promotion of both crops has been a key component of the anti-drug government-led policy, which has been implemented across the region since the 2000s. It is also implemented through a number of ongoing conservation initiatives aiming to upscale the adoption of sustainable practices in the production of both crops, given their relevant role in driving land cover changes across the region[53,54]. The resulting cluster of such initiatives in San Martin may, in turn, raise profitability challenges, as was experienced by some social enterprises in 2024 during the vertiginous increase in cocoa prices and ferocious competition between intermediaries dealing with higher operating costs in comparison to private companies, large cooperatives, and local intermediaries working in the same area. Meanwhile, the production of higher-quality yet less productive varieties of cacao and coffee among farmers in Quillabamba and Selva Central is more frequent due to the presence of relatively more favourable ecological conditions for their adoption. However, strong limitations faced by farmers to access specialized market niches for such varieties also explain well-being levels in these sites, which in average are lower than in San Martin[55,56].

In addition, in line with previous research[11,29,47], our findings also point to considerable well-being differences in association with demographic factors such as farmers' age and sex. In particular, older and female farmers reported less well-being across all farmer groups (Table S2-Supplementary Information). Further, according to our empirical results, the strategies assessed in our study made negligible contributions to the well-being contributions of older and female farmers (see Table S3-Supplementary Information). This finding reflects the primary orientation of value chain strategies toward younger and male farmers, who appear to have better access to inputs, labour, and the credits required to achieve productivity increases[47], as targeted by most of the corporate programmes and cooperatives included in our analysis. These results highlight the need for better and more inclusive tailoring of sustainability governance strategies to account for such demographic and contextual factors.

### Limitations

Two caveats should be kept in mind when interpreting our findings: First, the results are derived from a cross-sectional dataset, so the assessed strategies' apparent links with farmer well-being are correlations and not necessarily causal. Our research compares farmer well-being under different sustainability governance strategies and discusses plausible reasons for the associations between the strategies and well-being. Second, our results may not be directly applicable to other contexts. Future research should systematically compare the possible well-being contributions of corporate sustainability programmes, social enterprises, and cooperative strategies in other geographical contexts with different crops. This would improve the understanding of the external validity of the results presented here.

## Conclusions

This systematic comparison of well-being under corporate sustainability programmes, cooperatives, and social enterprises suggests that sustainability governance does indeed shape farmers' well-being. However, the decisive influence of other demographic, household, and contextual factors cannot be ruled out. Our results point to the importance of more localized, socially customized approaches to enhancing farmers' well-being in global value chains. Traditional approaches aimed at increasing productivity — often incorporating some agroecological practices — have been widely implemented across these value chains. The underlying assumption is that improvements in yields or crop quality will lead to financial gains, which will in turn positively impact farmers' well-being. However, complex difficulties, including limited land endowments, aging farmers, and gender marginalization, complicate the picture in rural contexts such as ours. They often challenge assumptions made by traditional value chain strategies, undermining their ability to enhance farmers' well-being[57]. Our empirical findings show that sustainability strategies with tailored approaches to address these challenges are associated with higher well-being outcomes among participating farmers; however, these strategies are difficult to scale up due to challenges such as increased implementation costs. Against this background, we agree with recent calls[58,59] for better interventions supported by a broader coalition of stakeholders, including private companies, governments, civil society, and ultimately final consumers.

## Methods

### Sampling

From June to September 2024, we conducted a total of 634 surveys. These encompassed farmers affiliated with 18 organizations involved in coffee and cocoa value chains across the Peruvian Amazon region, purposely selected according to the following criteria: (1) the organization pursues one of three assessed strategies: corporate sustainability programme, cooperative, or social enterprise; (2) the set of organizations provides a balanced representation of diverse value chain actors present (size, value chain position) as well as crop grown (coffee or cocoa); (3) stable organizational setting over multiple years; and (4) expressed willingness to participate in the study. In addition to these 18 organizations and the corresponding farmers. In addition, a group of 89 independent farmers distributed uniformly among the three sites (San Martin, Selva Central, and Quillabamba) was also surveyed. They regularly supply local intermediaries, and taken together, these independent farmers served as the reference group for comparisons. The first, fourth, and fifth author of this study directly coordinated with representatives of the selected organizations and private companies to facilitate their participation in the study via in-person meetings, emails, and phone calls between April and June 2024. Table S4 in the Supplementary Information provides an overview of the main features of each organization included in this study.

For each of the selected organizations, between three and 41 farmers (median = 30) were surveyed, mainly depending on the overall number of producers affiliated with the respective organization. In most cases, we referred to organizations' official lists of supplying or affiliated producers in order to obtain information about farmers' names and the villages where they reside. In a few cases, we used information directly provided by field technicians working in the study areas to identify and contact affiliated farmers. While this procedure was the most efficient way to identify and contact farmers to participate in our study, it could introduce selection bias. To assess this potential bias, we compared the characteristics of our sample with those of farmers from regions/study sites included in the most recent 2023 National Agrarian Survey. Similarities in terms of annual production, crop extension, and demographic variables (household members, gender, age) point to good representativeness present in our sample (Table S5 of Supplementary Information). Light distinctions of our sample include a relatively large share of farmers with completed basic education and managing smaller lands.

### Survey design and implementation

We designed and implemented a household survey containing six modules on demographic features, land use practices, income sources, self-reported well-being, farmers' assessment of their organization's services, and vulnerability to shocks. Surveys were conducted by two teams of enumerators: the first team comprised five members, including a field research supervisor and four enumerators who covered the San Martin site; the second comprised four members, including the field supervisor, and covered both the Selva Central and Quillabamba sites. The first author supervised the two teams and the entire data collection process. In both teams, surveys were mainly conducted by locally hired enumerators with experience in the coffee and cocoa sectors of the study are. Before the fieldwork started, the enumerators were trained in two three-day workshops led by the first author that focused on developing their capacities in the collection of digital surveys using the Open Data Kit (ODK) software for tablets. All but two surveys were conducted in-person, and most took 40–75 minutes for respondents to complete.

We employed the PWI[34] to assess well-being. A 0–10 Likert scale was used to capture responses to the question '*How satisfied are you regarding [the PWI domain]*' for each of the eight PWI domains, with 0 corresponding to '*not satisfied at all*' and 10 corresponding to '*completely satisfied*'. Importantly, we also assessed farmers' satisfaction in the domain of spirituality, given the relevance of this aspect to farmers' subjective well-being as observed by the first, third, and fifth authors during fieldwork, and findings from a previous analysis framed in a cultural context close to our study areas[50]. In addition, we included a question on how respondents perceived the relative importance of each well-being domain for use in calculating their subjective well-being. Finally, we included some questions adapted from standardized national questionnaires (e.g., National Agricultural Census) in our relevant survey modules in order to collect demographic and household data for statistical analyses. The final version of the survey was defined and installed in the fieldwork tablets after carrying out two pilot survey rounds in June 2024, in which 30 surveys were completed in villages near the study areas. The pilot-survey responses were not used in statistical analyses.

The ethical protocol applicable to this research was reviewed and approved by the European Research Council (ERC) and the host organization, Centre for Development and Environment (CDE) – University of Bern (Ethics Approval, 13 Oct. 2020). The activities involving this research were conducted in accordance with the local legislation and institutional requirements. Research participants were enrolled on a voluntary basis, and all participants provided their prior voluntary informed consent in written form. Our ethics protocol incorporated 'I do not know' and 'Refuse to answer' as valid responses to questions that respondents did not want to answer, and these were excluded from our statistical analyses.

### Data analysis

Following the standard procedure[34], we calculated the PWI for each farmer by summing their self-reported scores for each of the eight assessed well-being domains and then estimated the weighted PWI average among comparison groups. The average weights for each domain estimated from the overall sample were used in the calculation of individual PWIs according to the following formula:

$$PWI_i = \left( \frac{\sum_{n=1}^{k} s_{in} \cdot \overline{w_n}}{k} \right) \times 10 \qquad (1)$$

where: $PWI_i$ is the weighted Personal Well-being Index estimated for the $i$-th farmer adjusted to a 0–100 scale for comparability with similar studies[60]; $s_{in}$ is the score assigned by the farmer $i$ to the well-being domain $n$; $\overline{w_n}$ is the overall sample average weight for the well-being domain $n$; $k$ is the total number of assessed well-being domains. To assess differences among

sustainability strategies regarding composite PWI and each individual domain, we performed mean intergroup comparisons using the non-parametric Wilcoxon rank sum test for unpaired samples with Holm correction due to its flexibility in relation to the assumed normal distribution in samples. In addition, considering the original distribution of the PWI (0–100), we calculated the outcome according to a 0–1 scale by dividing it by 100, and then ran a generalized linear model (GLS) linked to a logit distribution of the following form to identify the major factors influencing its variance, including farmers' participation in sustainability strategies:

$$PWI_i/100 = \alpha + \beta_1 Strategies_i + \gamma_1 X_i + \delta_s + \varepsilon_{is} \qquad (2)$$

where: $Strategies_i$ is the categorical variable reflecting participation of the $i$-th farmer in one of the assessed sustainability strategies; $X_i$ is a vector of household-level variables including relevant demographic features (educational level, gender, number of household members), managed land sizes, agricultural revenues, and a binary variable reflecting strong trade relationships with their organization or company; $\delta_s$ is a vector of site fixed effects used to capture contextual differences among study sites; and $\varepsilon_{is}$ is the error term. The GLS model was applied to the overall sample comprising 599 farmers whose survey responses were completed, and then also to subsets of this, which were defined according to farmers' managed land size and main crop to capture heterogeneous effects across specific groups. Standard errors of all model coefficients were clustered at the case level to deal with potential autoregressive issues. Average marginal changes derived from regression results are reported within the main manuscript to facilitate interpretation of our estimations. Finally, graphical representations of our results were elaborated using packages from R-Studio (*ggplot2*, *ggpubr*) and Stata v. 18.5 (*coefplot*).

### Reporting summary

Further information on research design is available in the Nature Research Reporting Summary linked to this article.

### Data availability

Data needed to replicate our results are available at the BORIS repository (https://boris-portal.unibe.ch/handle/20.500.12422/220679)[61].

### Code availability

All the codes required to replicate our results are available at the BORIS repository (https://boris-portal.unibe.ch/handle/20.500.12422/220679)[61].

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

## Acknowledgements

We acknowledge funding from the European Research Council (ERC) under the European Union's Horizon 2020 research and innovation programme (COMPASS project, Grant agreement No. 949852). We thank all members of our local research teams in Quillabamba and Selva Central (Estrella Masías, Amanda Serrano, and Heberson Bustos) and San Martin (Marx Aguilar, José Pinedo, and José Ventura) for their enthusiasm and responsibility during fieldwork. We also thank Nicolas Porchet, Jimena Solar Álvarez, and Vincent Aggrey from the CDE at the University of Bern for comments and suggestions. Finally, we appreciate the feedback received from the journal's editors and three anonymous reviewers who helped us to improve the manuscript. All remaining errors are our own.

## Author contributions

J.G.M.Z. and C.O.: conceptualized the study; J.G.M.Z. designed data collection instruments, led data collection, performed statistical analyses, and wrote the first manuscript draft; C.O. supervised distinct stages of the research, including design of the survey instrument and data collection, and made major contributions to subsequent versions of the manuscript; R.B., S.B., and D.P.Z. contributed to data collection and improvements to the manuscript.

## Competing interests

The authors declare no competing interests.
