## [Transparent Peer Review file · Communications Sustainability]

Comparing corporate sustainability programmes, social entrepreneurship, and cooperatives in shaping farmers' well-being

Corresponding Author: Dr Javier Montoya-Zumaeta

Version 0:

Decision Letter:

Dear Dr Montoya-Zumaeta,

Your manuscript titled "Paths to prosperity? Well-being across corporate sustainability programmes, cooperatives, and social enterprises" has now been seen by 3 reviewers, and we include their comments at the end of this message. They find your work of interest, but some important points are raised. We are interested in the possibility of publishing your study in *Communications Sustainability*, but would like to consider your responses to these concerns and assess a revised manuscript before we make a final decision on publication.

We therefore invite you to revise and resubmit your manuscript, along with a point-by-point response that takes into account the points raised. Please highlight all changes in the manuscript text file.

For publication in *Communications Sustainability*, we request that you (1) outline your method and data in detail regarding farmers, sustainability certifications, and social enterprises, and (2) add qualitative analysis and deeper insight into findings based on organization characteristics.

Please submit your point-by-point responses as a separate file, distinct from your cover letter where you can add responses to the Editors' comments that you do not want to be made available to the reviewers. Word files are preferred. We recommend that any figures, tables or graphs that are included in the response to reviewers are also included in the main article or Supplementary Information.

Please use the following link to submit your revised manuscript, point-by-point response to the referees' comments (which should be in a separate document to any cover letter), a tracked-changes version of the manuscript (as a PDF file) and the completed checklist:

Link Redacted

We hope to receive your revised paper within six weeks; please let us know if you aren't able to submit it within this time so that we can discuss how best to proceed. If we don't hear from you, and the revision process takes significantly longer, we may close your file. In this event, we will still be happy to reconsider your paper at a later date, as long as nothing similar has been accepted for publication at *Communications Sustainability* or published elsewhere in the meantime.

Please do not hesitate to contact us if you have any questions or would like to discuss these revisions further. We look forward to seeing the revised manuscript and thank you for the opportunity to review your work.

Best regards,

Chenchen Ren, PhD
Editorial Board Member
Communications Sustainability
orcid.org/0000-0003-3308-2447

Martina Grecequet, PhD
Consulting Editor,
Communications Sustainability
Senior Editor,
Communications Earth & Environment

EDITORIAL POLICIES AND FORMATTING

- Behavioural and social science
- Ecological, evolutionary & environmental sciences
- Life sciences

Furthermore, please align your manuscript with our format requirements, which are summarized on the following checklist: <https://www.nature.com/documents/commsj-phys-style-formatting-checklist-article.pdf> Communications Sustainability formatting checklist

and also in our style and formatting guide <https://www.nature.com/documents/commsj-phys-style-formatting-guide-accept.pdf> Communications Sustainability formatting guide .

***** DATA:** Communications Sustainability endorses the principles of the Enabling FAIR data project (<http://www.copdess.org/enabling-fair-data-project/>). We ask authors to make the data that support their conclusions available in permanent, publicly accessible data repositories. (Please contact the editor if you are unable to make your data available).

All Communications Sustainability manuscripts must include a section titled "Data Availability" at the end of the Methods section or main text (if no Methods). More information on this policy, is available at <http://www.nature.com/authors/policies/data/data-availability-statements-data-citations.pdf>.

If a community resource is unavailable, data can be submitted to generalist repositories such as <https://figshare.com/> or <http://datadryad.org/> Dryad Digital Repository. Please provide a unique identifier for the data (for example a DOI or a permanent URL) in the data availability statement, if possible. If the repository does not provide identifiers, we encourage authors to supply the search terms that will return the data. For data that have been obtained from publicly available sources, please provide a URL and the specific data product name in the data availability statement. Data with a DOI should be further cited in the methods reference section.

REVIEWER COMMENTS:

Reviewer #1 (Remarks to the Author):

The paper claims to critically examine the relative contributions of different strategies for the governance of sustainability in agri-food systems using a case study of coffee and cocoa growing farmers in Peru. While there are studies examining the impacts of social enterprises (Misunguzi et al., 2023), corporate sustainability programmes (Misra et al., 2024), and cooperative-based strategies (Abibawe and Haile ., 2013), there are relatively fewer studies that compare the impacts. The paper effectively compares these impacts using the Personal Well-being index, providing insights into how these impacts differ and accounting for socioeconomic heterogeneity within the population. Overall, this study is insightful; however, I found that there are areas for improvement. 1) There is tremendous diversity in how these programs are implemented in different regions; however, the authors fail to provide this context in the case study. The paper does not provide details on the different types of programs, making it difficult to discern the drivers of differences in outcomes. 2) The authors are over-reliant on correlational statistics to prove their main points. I would prefer if the authors unpack the findings based on the features of the organizations that they are studying 5) Finally, given the relatively small sample size, lack of panel data, and small counterfactual (~30 farmers), it would be more convincing if the authors presented the study as a case study providing more qualitative analysis.

Reviewer #2 (Remarks to the Author):

Thank you for this article. It is well written and interesting and addresses an important topic. I suggest only minor changes. I am surprised by the location of the methods section after the conclusions. There is important information here that the reader wants when interpreting the results. I suggest moving it before the results section.

Are there no cases of farmer in SE or CSR who are also members of cooperatives?

What sustainability certifications do the farmers participate in? Schemes have different emphases and we would expect different impacts on well-being.

Line 39—I think we need a little more information about social enterprises—how do they purport to create change/sustainability. You mention “purpose over profit.” Do you mean corporate profit? Don’t they attempt to increase farmer profit?

Line 42—in your discussion of cooperatives you should mention the principle of concern for the community as a way cooperatives seek to improve the well-being of members and communities.

Line 344 Groups of independent farmers? Are they members of farmer groups that are not cooperatives? Or completely independent?

Reviewer #3 (Remarks to the Author):

Thank you for the opportunity to review the manuscript titled "paths to prosperity? well-being across corporate sustainability programmes, cooperatives, and social enterprises". Overall, I found the manuscript to be of exceptional quality. It is well written, theoretically-informed and empirically-rich.

My recommendation is to publish the manuscript with very minor revisions.

The first suggestion relates to the section titled "the influence of demographic and social-ecological context factors". While I appreciate the rich discussion on crop type, I was wanting to know just a bit more on the broader environmental factors--both drivers/inputs of the farming systems and the consequences associated with corporate sustainability/cooperatives vs. independent production. I don't think much is needed here. A sentence or two should suffice to describe why the results from this region in Peru might be specific to the ecological context and potentially also generalizable beyond the region as well.

The second suggestion relates to this point. The author(s) mention the importance of policies. While I realize this is a bit beyond the scope of the paper, I do believe it would be helpful to at least mention one or two of the key policies and/or political context that might influence the results of the study. I found myself wondering how the study would relate to a neighboring country (e.g., Colombia or Brazil) where production systems might be similar but the policy environment is quite different. Again, I realize this is a bit beyond the scope of the manuscript, but a few sentences on this point would help limit the readers' imaginations.

Beyond that, I want to reiterate my extremely positive view on the quality of this manuscript. It will be an important contribution to the overall literature.

** Visit Nature Portfolio's author and referees' website at www.nature.com/authors for information about policies, services and author benefits**

Communications Sustainability is committed to improving transparency in authorship. As part of our efforts in this direction, we are now requesting that all authors identified as 'corresponding author' create and link their Open Researcher and Contributor Identifier (ORCID) with their account on the Manuscript Tracking System prior to acceptance. ORCID helps the scientific community achieve unambiguous attribution of all scholarly contributions. You can create and link your ORCID from the home page of the Manuscript Tracking System by clicking on 'Modify my Springer Nature account' and following the instructions in the link below. Please also inform all co-authors that they can add their ORCIDs to their accounts and that they must do so prior to acceptance.

Version 1:

Decision Letter:

Dear Dr Montoya-Zumaeta,

Your manuscript titled "Paths to prosperity? Well-being across corporate sustainability programmes, cooperatives, and social enterprises" has now been seen by our reviewers, whose comments appear below. In light of their advice we are delighted to say that we are happy, in principle, to publish a suitably revised version in Communications Sustainability.

We therefore invite you to edit your manuscript to comply with our format requirements and to maximise the accessibility and therefore the impact of your work.

EDITORIAL REQUESTS:

****Please take care to match our formatting and policy requirements. We will check revised manuscript and return manuscripts that do not comply. Such requests will lead to delays. ****

SUBMISSION INFORMATION:

OPEN ACCESS:

Communications Sustainability is a fully open access journal. Articles are made freely accessible on publication. For further information about article processing charges, open access funding, and advice and support from Nature Portfolio, please visit <https://www.nature.com/commssustain/open-access>

Link Redacted

Best regards,

Chenchen Ren, PhD
Editorial Board Member
Communications Sustainability
orcid.org/0000-0003-3308-2447

Martina Grecequet, PhD
Consulting Editor,
Communications Sustainability
Senior Editor,
Communications Earth & Environment

REVIEWERS' COMMENTS:

Reviewer #1 (Remarks to the Author):

Thank you for your efforts to improve the manuscript. I think it has improved significantly and is almost ready for publication. I would only suggest that you move Supplementary Table 4 to the main body of the manuscript.

Reviewer #2 (Remarks to the Author):

Thank you for your revisions. I think that you have addressed all of the reviewer comments. I recommend this manuscript for publication.

Reviewer #3 (Remarks to the Author):

I was pleased to see the well thought-out revisions and rebuttal/response to the reviews. I recommend the article to be published as is.

** Visit Nature Portfolio's author and reviewers' website at <http://www.nature.com/authors> for information about policies, services and author benefits**

Paths to prosperity? Well-being across corporate sustainability programmes, cooperatives, and social enterprises

Rebuttal letter to reviewers

In the following table, we use the left column to reproduce each reviewer’s comment and the right column to explain how we address these in the revised manuscript.

Reviewers’ comments	Authors’ response
Editor’s comments: For publication in Communications Sustainability, we request that you (1) outline your method and data in detail regarding farmers, sustainability certifications, and social enterprises, and	Thank you for securing the reviews, which have improved the quality of the manuscript. As requested, we have provided clearer and more detailed information in relation to farmers, sustainability certifications, and social enterprise also addressing reviewers’ comments in this regard. They are now mentioned more explicitly in the Introductory and Discussion Sections of the main manuscript as well as in the Supplementary Material (Table S4).
(2) add qualitative analysis and deeper insight into findings based on organization characteristics.	We provide more information connecting findings and organization characteristics also addressing second and third comments of the Reviewer 1 particularly within the subsection entitled “Comparing the well-being contributions of corporate programmes, cooperatives, and social enterprises”
Reviewer #1 (Remarks to the Author): The paper claims to critically examine the relative contributions of different strategies for the governance of sustainability in agri-food systems using a case study of coffee and cocoa growing farmers in Peru. While there are studies examining the impacts of social enterprises (Misunguzi et al., 2023), corporate sustainability programmes (Misra et al., 2024), and cooperative-based strategies (Abibawe and Haile ., 2013), there are relatively fewer studies that compare the impacts. The paper effectively compares these impacts using the Personal Well-being	Thank you, we appreciate this largely positive impression of our manuscript as well as the constructive suggestions for improvement.

index, providing insights into how these impacts differ and accounting for socioeconomic heterogeneity within the population. Overall, this study is insightful; however, I found that there are areas for improvement.	
1) There is tremendous diversity in how these programs are implemented in different regions; however, the authors fail to provide this context in the case study. The paper does not provide details on the different types of programs, making it difficult to discern the drivers of differences in outcomes.	Thanks for this comment. We added Table S4 in the Supplementary Material to present the characteristics of each corporate sustainability programme, social enterprise and cooperative included in our study. This overview illustrates the specific organizational characteristics that we cover in the Results and Discussion section to supports the points we made about farmers' wellbeing differences among the sustainability strategies. Moreover, we further contextualize the cases by elaborating more on the various strategies in the Introduction section, within the confines of the word limit of this journal article.
2) The authors are over-reliant on correlational statistics to prove their main points. I would prefer if the authors unpack the findings based on the features of the organizations that they are studying.	Thank you for this comment. We explicitly discuss the point made here as one of the limitations of our study indicating that although we do not claim that analysed sustainability strategies caused differences on farmers' wellbeing, our approach allows us to identify and discuss "plausible reasons for the associations between the strategies and well-being" (lines 327-328). Furthermore, through Table S4, we provide more information that support the points we raise for explaining farmers' wellbeing differences across sustainability strategies by connecting these to features of all organizations included in the study (cf. response to comment 1 above).
3) Finally, given the relatively small sample size, lack of panel data, and small counterfactual (~30 farmers), it would be more convincing if the authors presented the study as a case study providing more qualitative analysis.	Thank you for this comment. In relation to this comment is important to clarify: 1) Our control group is comprised by 90 (not 30) independent farmers as better explained now in the revised manuscript (lines 360-363), and 2) our findings rely on analysing 18 study cases (i.e. organizations), rather than 1.

	We explicitly discuss the limitations of our study indicating that although we do not claim that analysed sustainability strategies caused differences on farmers' wellbeing, our approach allows us to discuss "... plausible reasons for the associations between the strategies and well-being" (lines 327-328).
Reviewer #2 (Remarks to the Author): Thank you for this article. It is well written and interesting and addresses an important topic. I suggest only minor changes.	Thank you. We appreciate your positive overall impression of our manuscript as well as the constructive suggestions.
I am surprised by the location of the methods section after the conclusions. There is important information here that the reader wants when interpreting the results. I suggest moving it before the results section.	The location of the Methods Section corresponds to the style and format required by Nature's Communications journals according to their guide, which can be consulted here. It is for this reason that we opted to maintain the Methods section in the same location.
Are there no cases of farmer in SE or CSR who are also members of cooperatives?	As explained in the Methods Section (lines 369-373), we first relied on field technicians and managers of each organization we worked with (including cooperatives and companies) to identify farmers. Once identified, we confirm their affiliation after directly asking identified farmers whether they regularly provide cacao/coffee to the respective organization and whether they have received any benefit from this. Indeed, we found a few former members of cooperatives who are currently engaged with SE (social enterprises) or CSP (corporate sustainability programmes). When conducting surveys with them, our enumerators were trained to clarify that questions related to the organizations' services and benefits refer to the company they currently work with rather than previous ones.
What sustainability certifications do the farmers participate in? Schemes have different emphases and we would expect different impacts on well-being.	After screening organizational-level information collected previously within the frame of our research project, we identified Organic, Fair Trade and Rainforest as the most frequently used certifications schemes, consistent with global trends in cocoa and

	coffee. This information is now explicitly mentioned in the revised manuscript (line 222). While we acknowledge that the thoughts on different certification schemes are relevant for science and policy, this study is among the first to disentangle wellbeing contributions from different strategies of sustainability governance (i.e. CSP, SE, CO), with some of them presenting a strong correlation with the adoption of third-party certifications as in the case of cooperatives. We discuss this overlap in the Results and Discussion Sections (see lines 168-171, 226-232). To address deeper the question about the impacts of the diverse certification schemes would require another research design and therefore exceeds the scope of this manuscript.
Line 39—I think we need a little more information about social enterprises—how do they purport to create change/sustainability. You mention “purpose over profit.” Do you mean corporate profit? Don’t they attempt to increase farmer profit?	As suggested, we have clarified: “Social enterprises take a business approach to attain a social and environmental purpose. They often aim to create change by prioritizing this corporate purpose over profit in decision-making, with the former enshrined in corporate mission and accountability structures. Fair prices, direct trade relations and risk sharing are key instruments of many social enterprises in the agri-food sector.” (lines 39-43)
Line 42—in your discussion of cooperatives you should mention the principle of concern for the community as a way cooperatives seek to improve the well-being of members and communities.	Thank you for this comment. We now have added in the Introduction: “Cooperatives, in particular, often surge from existing communal organizations and attempt to incorporate, disseminate and strengthen collective values across its members as a pathway to enhance their wellbeing while maintaining roots with communities.” (lines 64-66).
Line 344 Groups of independent farmers? Are they members of farmer groups that are not cooperatives? Or completely independent?	They are independent farmers in the sense that they work independently and sell cacao or coffee to the buyer who offer higher prices and better commercial conditions, often at local warehouses. Although some of them repeatedly sell to a local intermediate, these relationships are not based on formal agreements but largely on the market

	conditions in place for the traded crop within the sourcing region.
Reviewer #3 (Remarks to the Author): Thank you for the opportunity to review the manuscript titled "paths to prosperity? well-being across corporate sustainability programmes, cooperatives, and social enterprises". Overall, I found the manuscript to be of exceptional quality. It is well written, theoretically-informed and empirically-rich. My recommendation is to publish the manuscript with very minor revisions.	Thank you. We appreciate your very positive overall assessment of our manuscript and the suggested improvements.
The first suggestion relates to the section titled "the influence of demographic and social-ecological context factors". While I appreciate the rich discussion on crop type, I was wanting to know just a bit more on the broader environmental factors--both drivers/inputs of the farming systems and the consequences associated with corporate sustainability/cooperatives vs. independent production. I don't think much is needed here. A sentence or two should suffice to describe why the results from this region in Peru might be specific to the ecological context and potentially also generalizable beyond the region as well.	Thank you for this comment. In the Section you refer to, we have added : “This reflects differences among these sites in terms of market accessibility, ecological and institutional factors as we discuss further in the next Section” (lines 182-184), and then in the Discussion Section we added: “Meanwhile, the production of higher quality yet less productive varieties of cacao and coffee among farmers in Quillabamba and Selva Central is more frequent due to the presence of relatively more favourable ecological conditions for their adoption. However, strong limitations faced by farmers to access specialized market niches for such varieties also explain well-being levels in these sites that in average are lower than in San Martin.” (lines 308-312) Regarding generalizability, we expect that the results in the comparison of CSP, SE and CO are valid beyond our study region, but future research will need to test this. In particular, the insight on the importance of tailoring the strategies to socio-economic, institutional and ecological factors is key, and our empirical results demonstrate the factors that proved statistically significant in our study context – this is crucial for tailoring sustainability strategies to their particular contexts in different regions.
The second suggestion relates to this point. The author(s) mention the importance of policies. While I realize this is a bit beyond	Thanks for this comment. In this regard, we have added in the Discussion Section: “Notably, the promotion of both crops has

the scope of the paper, I do believe it would be helpful to at least mention one or two of the key policies and/or political context that might influence the results of the study. I found myself wondering how the study would relate to a neighboring country (e.g., Colombia or Brazil) where production systems might be similar but the policy environment is quite different. Again, I realize this is a bit beyond the scope of the manuscript, but a few sentences on this point would help limit the readers' imaginations.	been a key component of the anti-drug government-led policy, which has been implemented across the region since the 2000s. It is also implemented through a number of ongoing conservation initiatives aiming to upscale the adoption of sustainable practices in the production of both crops, given their significant role to drive land cover changes across the region.” (lines 300-304).
Beyond that, I want to reiterate my extremely positive view on the quality of this manuscript. It will be an important contribution to the overall literature.	Again, we thank you for the positive appreciation of our manuscript.

Comparing corporate sustainability programmes, social entrepreneurship, and cooperatives in shaping farmers' well-being
(COMSSUSTAIN-25-0116A)

Rebuttal letter to reviewers (10.10.2025)

In the following table, we use the left column to reproduce each reviewer's comment and the right column to explain how we address these in the revised manuscript.

Reviewers' comments	Authors' response
Reviewer #1 (Remarks to the Author): Thank you for your efforts to improve the manuscript. I think it has improved significantly and is almost ready for publication. I would only suggest that you move Supplementary Table 4 to the main body of the manuscript.	Thank you very much for your positive evaluation to our manuscript. In relation to your suggestion, after a thoughtful process we have decided to maintain the Table S4 as Supplementary Material because it presents contextual information that is not the main focus of this manuscript and due to its size.
Reviewer #2 (Remarks to the Author): Thank you for your revisions. I think that you have addressed all of the reviewer comments. I recommend this manuscript for publication.	Thank you very much for your positive evaluation to our manuscript.
Reviewer #3 (Remarks to the Author): I was pleased to see the well thought-out revisions and rebuttal/response to the reviews. I recommend the article to be published as is.	Thank you very much for your positive evaluation to our manuscript.